

# Long-term passage impacts human dental pulp stem cell activities and cell response to drug addition *in vitro*

Somying Patntirapong[1], Juthaluck Khankhow[2] and Sikarin Julamorn[2]

[1] Thammasat University Research Unit in Dental and Bone Substitute Biomaterials, Faculty of Dentistry, Thammasat University, Pathumthani, Thailand
[2] Faculty of Dentistry, Thammasat University, Pathumthani, Thailand

## ABSTRACT

**Background**. Dental pulp stem cells (DPSCs) possess mesenchymal stem cell characteristics and have potential for cell-based therapy. Cell expansion is essential to achieve sufficient cell numbers. However, continuous cell replication causes cell aging *in vitro*, which usually accompanies and potentially affect DPSC characteristics and activities. Continuous passaging could alter susceptibility to external factors such as drug treatment. Therefore, this study sought to investigate potential outcome of *in vitro* passaging on DPSC morphology and activities in the absence or presence of external factor.

**Methods**. Human DPSCs were subcultured until reaching early passages (P5), extended passages (P10), and late passages (P15). Cells were evaluated and compared for cell and nuclear morphologies, cell adhesion, proliferative capacity, alkaline phosphatase (ALP) activity, and gene expressions in the absence or presence of external factor. Alendronate (ALN) drug treatment was used as an external factor.

**Results**. Continuous passaging of DPSCs gradually lost their normal spindle shape and increased in cell and nuclear sizes. DPSCs were vulnerable to ALN. The size and shape were altered, leading to morphological abnormality and inhomogeneity. Long-term culture and ALN interfered with cell adhesion. DPSCs were able to proliferate irrespective of cell passages but the rate of cell proliferation in late passages was slower. ALN at moderate dose inhibited cell growth. ALN caused reduction of ALP activity in early passage. In contrast, extended passage responded differently to ALN by increasing ALP activity. Late passage showed higher collagen but lower osteocalcin gene expressions compared with early passage in the presence of ALN.

**Conclusion**. An increase in passage number played critical role in cell morphology and activities as well as responses to the addition of an external factor. The effects of cell passage should be considered when used in basic science research and clinical applications.

Corresponding author
Somying Patntirapong,
psomying@tu.ac.th

## INTRODUCTION

Mesenchymal stem cells (MSCs) are multipotent cells, capable of giving rise to many cell lineages. Although MSCs were primarily identified in the bone marrow, they can be isolated from tissues in the oral cavity. Human dental pulp stem cells (DPSCs) are post-natal populations of MSCs residing in the pulp cavity of permanent teeth (*Tatullo et al., 2015*). DPSCs are considered feasible and promising source of autologous stem cells because they have MSC qualities (*Tatullo et al., 2015*; *Awais et al., 2020*; *Al Madhoun et al., 2021*) and are cost-effective. Their isolation procedure is less invasive and can be obtained from discarded or removed teeth such as premolar and third molar. Isolated *ex vivo* DPSCs are characterized as cells with a high level of clonogenicity and proliferation and share a similar immunophenotype to that of bone marrow MSCs *in vitro* (*Gronthos et al., 2000*). DPSCs, when placed under specific conditions, generate different cell lineages: odonto/osteogenic, chondrogenic, neurogenic, adipogenic, and myogenic (*Mangano et al., 2011*; *Kogo et al., 2020*). These cells are capable of forming mineralized nodule *in vitro* and regenerate a dentin-like structure *in vivo* (*Tatullo et al., 2015*).

Human stem cells can easily be cultivated, expanded, and cryopreserved as well as produce progeny with strong differentiation capacity. Therefore, use of these cells has become common for many purposes ranged from scientific studies to tissue engineering in order to replace damaged cells using autologous transplant in various diseases. DPSCs have also been increasingly studied and employed in regenerative field including cell-guided regeneration for correcting of bone defects (*d'Aquino et al., 2009*; *Mangano et al., 2011*; *Monti et al., 2017*; *Awais et al., 2020*).

MSC-based therapies and studies demand large scale *ex vivo*/*in vitro* expansion to reach the numbers required for cell therapy. Cell deterioration after prolonged expansion in cell culture is an unavoidable physiological consequence (*Hayflick & Moorhead, 1961*). Late cell passages affect cell appearance, proliferative capability, and osteogenic differentiation (*Yang et al., 2018*; *Grotheer et al., 2021*). MSCs gradually lose their typical fibroblast shape and lack morphological homogeneity (*Yang et al., 2018*). The rate of cell doublings significantly decreases (*Yang et al., 2018*), which are not suitable for therapeutic application. DPSCs undergoing many serial passaging also display a reduction in cell proliferation and viability (*Martin-Piedra et al., 2014*; *Yan et al., 2022*). *In vivo* transplantation of DPSCs demonstrates a restriction in the differentiation capacity into osteoblast lineage at high passage (9th) (*Yu et al., 2010*). Cell adhesion and spreading are crucial for cell proliferation, differentiation, and mineralization (*Geng et al., 2020*). Thus, the success of cell attachment and interaction with the surface of the substrates depend on these activities. Nevertheless, these cellular aspects as well as cell morphology of long-term cultivated DPSCs are still unclear.

DPSCs have ability to respond to several influences such as caries and other biochemical and mechanical factors. DPSCs respond to high dose of lipopolysaccharide by increase in cell death (*Gao et al., 2020*). On the other hand, *ex vivo* DPSCs exposure to deep caries still have proliferative capability and express higher angiogenic marker (*Chen et al., 2021*). Activation of $K^+$ channels in DPSCs induces the differentiation of DPSCs into neuron-like cells (*Kogo et al., 2020*). Long-term expansion could be an influence on cell response to an
external factor/chemical factor/inciting factor. Due to the lack of this information, it was thus essential to determine the influence of *in vitro* passaging on cellular qualities under an external condition. Therefore, the present work sought to investigate (1) DPSC activities at different passages to determine the optimal passage and (2) cell activities at different passages in the present of external factor. Alendronate (ALN) is a type of bisphosphonate (BP), which acts as an anti-resorptive drug. It is known to have inhibitory effects on osteoblasts (*Patntirapong, Singhatanadgit & Arphavasin, 2014*; *Patntirapong et al., 2021b*). BP mainly locates in alveolar bone; however, it is also found to deposit throughout the tooth, including the pulp chamber and the dentinoenamel junction (*Kozloff et al., 2009*). Treatment of BP impairs tooth formation (*Hasoon & AL-Ghaban, 2013*) and alters dental pulp cell activities in both odontoblasts and non-odontoblasts (*De Barros Silva et al., 2017*; *De Barros Silva et al., 2019*). Thus, BP could affect the dental pulp cells. In this study, ALN drug treatment was served as an external factor added to DPSC culture. Long-term DPSC subcultures from passage 5-15 under ALN-free and ALN conditions were evaluated for cell adhesion, cell morphology, cell proliferation, and alkaline phosphatase activity.

## MATERIALS AND METHODS

### Cell culture and treatments

The manuscript of this laboratory study has been written according to Preferred Reporting Items for Laboratory studies in Endodontology (PRILE) 2021 guidelines. Human dental pulp stem cells (DPSCs; Lonza, PT-5025) were implemented in this study because their lineage and species origin present a practical *in vitro* system for studies. They are commercially available primary cell line that can be used in follow-up and confirmatory experiments by our research team as well as other investigators (*Irfan et al., 2022*). According to the company's data, DPSCs are tested for $CD105^+$, $CD166^+$, $CD29^+$, $CD90^+$, $CD73^+$, $CD133^-$, $CD34^-$, $CD45^-$ using flow cytometry. Cells were continuously passaged until reaching passage 16. Cell passage 4–6, 9–11, and 14–16 were used in the experiments and these passages were referred to as P5 (early passage), P10 (extended passage), and P15 (late passage), respectively. DPSCs were maintained in standard culture media, which were Dulbecco's modified Eagle's medium (DMEM, Gibco) supplement with 10% fetal bovine serum and 1% penicillin/streptomycin at 37 °C and 5% $CO_2$ humidified atmosphere. DPSCs were plated at the density of 7,500 cells/cm$^2$ and then treated with ALN at various concentrations (0, 0.1, 0.5, 5, 10 $\mu$M). In this study, 0.1–0.5 $\mu$M ALN was considered low concentration and 5–10 $\mu$M ALN was moderate concentration. For cell differentiation, DPSCs were cultured under osteogenic media (OM), which were standard culture media supplemented with 50 $\mu$g/ml ascorbic acid (BDH), 10 mM $\beta$-glycerophosphate (Sigma-Aldrich, St. Louis, MO, USA), and 100 nM dexamethasone.

### Cell adhesion assay

Cells were seeded and treated with ALN for 5 h. Although cells can attach to the plate within minutes after cell seeding, DPSCs were incubated for longer periods to let the cells adequately contact the substrate. Thus, the cells had stronger adhesion strength and had enough numbers of cells to perform the assays and analysis. Non-adherent cells were

removed by gently washing with phosphate-buffered saline solution. Adherent cells were fixed with 4% paraformaldehyde for 15 min at room temperature. Cells were stained with 1% crystal violet (Reag. Ph. Eur.) for 20 min and rinsed carefully. Cells were examined under a microscope (Nikon Eclipse Ti, Nikon Instruments) at 100x magnification. Ninety-six images of cells from four wells were captured using NIS element AR 4.11.00 software. The numbers of cells ranged from 190-2084 were recorded and analyzed.

## Cell and nuclear morphological assay

Cells were treated with ALN for 3 days. Cells were fixed with 4% paraformaldehyde for 15 min and incubated with 1% crystal violet for 20 min. Cell appearance was monitored at 100x magnification. Forty areas from four wells were recorded and 410–600 cells were analyzed. Nuclear shape was stained with 4′,6-diamidino-2-phenylindole (DAPI) at the dilution 1:1000 for 5 min. Nuclei were visualized under a confocal microscope (Nikon Eclipse Ti, Nikon Instruments) at 200x magnification. The numbers of nuclei ranged from 447–675 were examined.

## Image analysis

Quantitative data was analyzed by ImageJ software version 1.53k Java 1.8.0 (National Institute of Health). Images of cells stained with crystal violet were assessed according to previous report (*Patntirapong, Charoensukpatana & Thaksinawong, 2022*). In brief, the scale was set in a micrometer unit. Original images were processed by automated detection mode. The background of images was eliminated using the Subtract Background function. Images were converted to 8-bit grayscale and were processed by Auto Threshold commands. Cluster cells were optionally segmented by Watershed function. Cells were identified and analyzed by Analyze Particles function. Data from isolated cells were collected. In the cell morphological test, the particles smaller than 200 $\mu m^2$ were excluded and identified as debris. For nuclear analysis, images of the nuclei were processed as described in previous report (*Patntirapong et al., 2021a*). Quantitative data such as area ($\mu m^2$), perimeter ($\mu m$), roundness, aspect ratio (AR), circularity, solidity, and number were measured.

## Cell proliferation assay

DPSCs treated for 1, 3, and 7 days were incubated with 10 $\mu L$ of the CCK-8 solution (Dojindo Laboratories) at 37 °C for 3 h according to company instruction. The plates were analyzed using a microplate reader (Sunrise) with Megellan software, V6.6 at the absorbance 450 nm.

## Protein measurement and alkaline phosphatase (ALP) activity

Cells were cultured in OM and treated with ALN for 7 days. Media were collected for measuring ALP released in the media. Cells were lysed with Triton X-100 lysis buffer (50 mM Tris, 150 mM NaCl, and 1% Triton X-100, pH 10). Cell lysates were measured for total proteins using the BCA protein assay kit (Pierce). The mixture was read at absorbance 562 nm using a microplate reader. Total protein was quantified against known BCA protein concentration. The aliquots with an equal amount of protein content from each sample and media were incubated with ALP substrate using ALP assay kit (Elabscience) at 37 °C

for 15 min. The optical density of p-nitrophenol was determined by a spectrophotometer at 520 nm. ALP activity was calculated relative to standard phenol solution and expressed as ng/ml.

### Real-time polymerase chain reaction (PCR)

mRNA was isolated from OM-induced cells using the Total RNA Mini kit (Geneaid). All purified mRNA samples were processed into cDNA using oligo dT (TAKARA BIO INC.). cDNA samples were amplified in a reaction mix containing KAPA SYBR® FAST PCR Kit Master Mix (Thermo Fisher Scientific, Waltham, MA, USA)) and the forward and reverse primer pair sequences (Sigma-Aldrich, St. Louis, MO, USA). The amplification was run in QuantStudio™ 3 Real-Time PCR Systems (Thermo Fisher Scientific). The cycles were set at 50 °C for 2 min initial heating, 95 °C for 1 min, 40 cycles of 95 °C for 30 s, followed by 60 °C for 30 s with 72 °C elongation for 30 s. The forward/reverse primer pairs were as follows: glyceraldehyde 3-phosphate dehydrogenase (GAPDH) "CTCATTTCCTGGTATGACACC" and "CTTCCTCCTGTGCTCTTGCT"; collagen type I (Col I) "TGACCTCAAGATGTGCCACT" and "ACCAGACATGCCTCTTGTCC"; osteocalcin (OC) "TCACACTCCTCGCCCTATTG" and "TCGCTGCCCTCCTGCTTG"; Bone sialoprotein (BSP) "AACCTACAACCCCACCACAA" and "AGGTTCCCCGTTCT-CACTTT"; dentin sialophosphoprotein (DSPP) "AGACGAGGGTTCTGGTGATG" and "TCTTCTTTCCCATGGTCCTG"; dentin matrix acidic phosphoprotein 1 (DMP1) "GCAGAGTGATGACCCAGAG" and "GCTCGCTTCTGTCATCTTCC". The gene copy number was normalized with GAPDH. Data were presented in fold changes relative to control of each group.

### Statistical analysis

Four independent experiments were performed. Data was tested for normal distribution using the Kolmogorov–Smirnov test (GraphPad Prism 9.4.0). Data that was normally distributed was analyzed by ANOVA followed by Dunnett's test. Data that was not normally distributed was analyzed by Kruskal-Wallis test followed by Dunn's procedure. Significance was assigned as * $p < 0.05$, ** $p < 0.01$, *** $p < 0.001$ vs P5 in the same ALN treatment group; [a] $p < 0.05$, [aa] $p < 0.01$, [aaa] $p < 0.001$ vs P5A0 in P5 group; [b] $p < 0.05$, [bb] $p < 0.01$, [bbb] $p < 0.001$ vs P10A0 in P10 group; [c] $p < 0.05$, [cc] $p < 0.01$, [ccc] $p < 0.001$ vs P15A0 in P15 group.

## RESULTS

### Alteration of cell morphology in early, extended, and late passages under ALN-free and ALN conditions

Microscopic images of DPSCs at P5, P10, and P15 are depicted in Figs. 1A, 1B, and 1C, respectively. Cells in each condition showed different cell size and shape. Untreated DPSCs at P5 were small in size and mainly spindle shape (Fig. 1Ai), which mostly maintained the shape and size as observed in P1 cells (Fig. 1D). Continuous culture led to morphological alteration. P10 and P15 cells gradually spread and appeared as polygonal shape. Cells displayed less homogenous morphologies. DPSCs at P15 noticeably exhibited enlarged

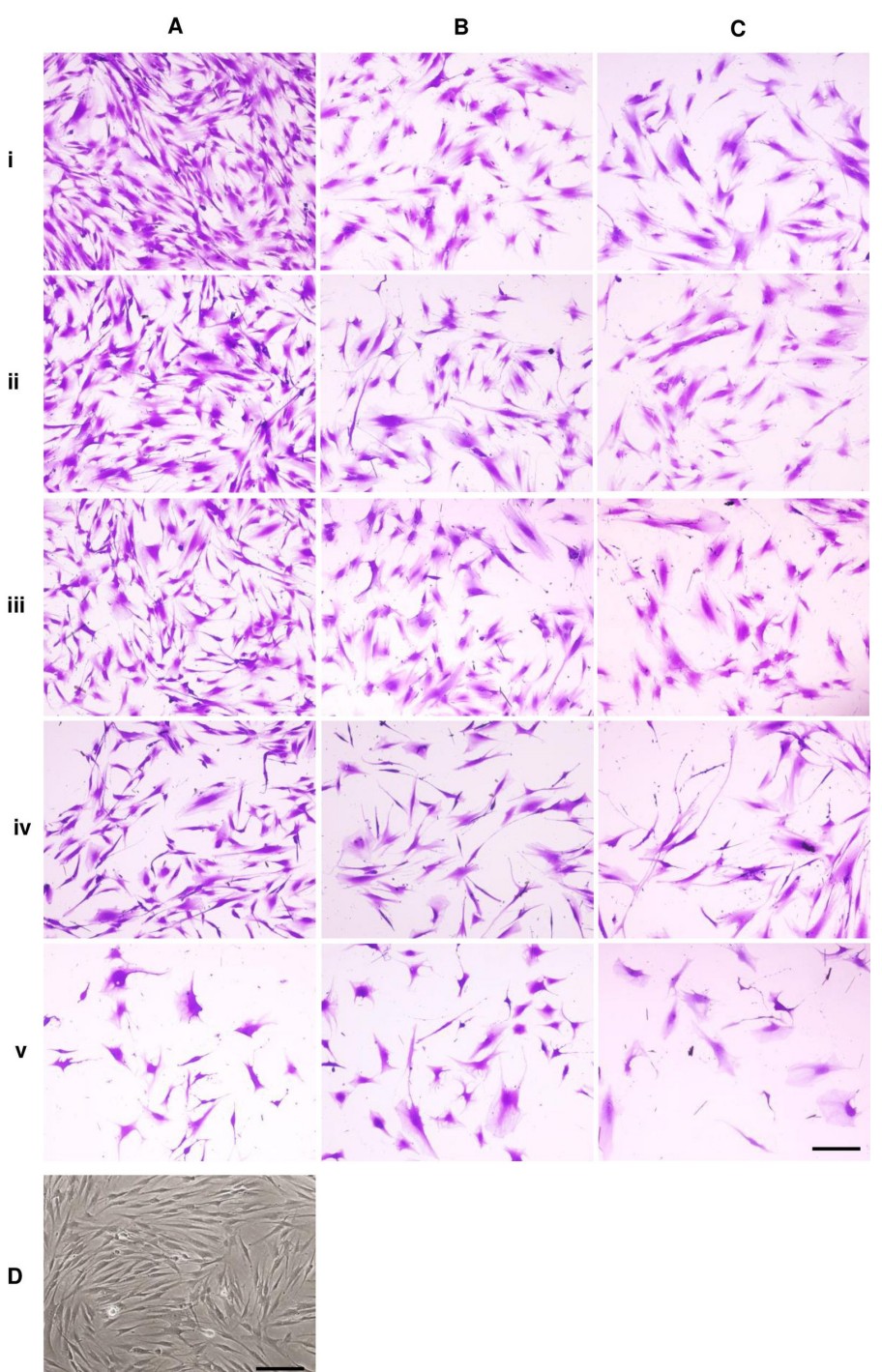

**Figure 1  Dental pulp stem cell morphology.**

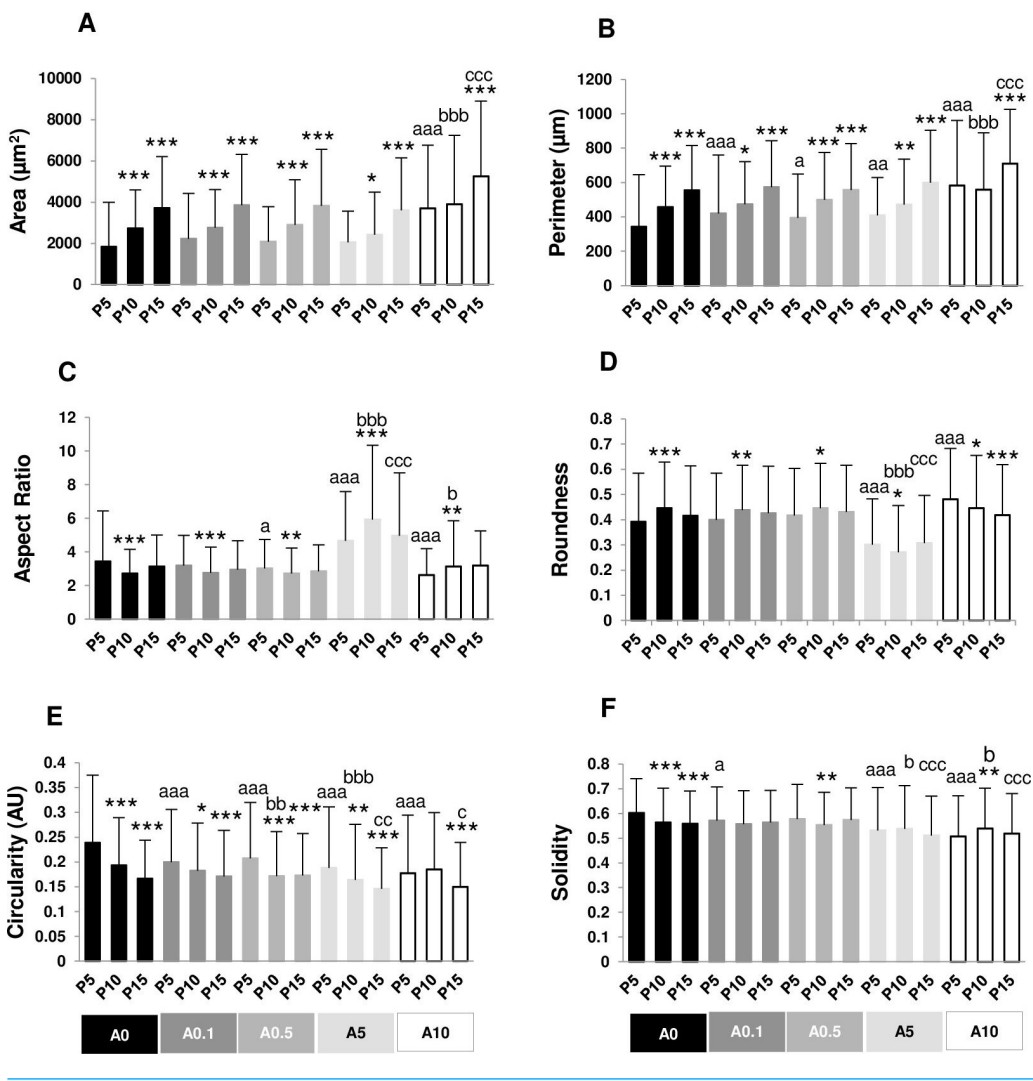

**Figure 2** (A–F) Measurement of cell morphology.

cell bodies with extended cellular processes. Fewer cells were observed for P10 and P15 (Figs. 1Bi and 1Ci). The addition of ALN to P5 cells altered the cell shape to fusiform (Fig. 1Aiv) or polygonal with more cellular processes (Fig. 1Av). However, cell shrinkage from 10 µM ALN (A10) was observed (Fig. 1Av). The addition of A5 and A10 to P10 and P15 also altered cell shape (Figs. 1B and 1C).

Cell area and perimeter of P10 and P15 were larger than those of P5 in ALN-free and ALN-treated groups (Figs. 2A and 2B). Treatment with A10 increased cell area in every passage (Fig. 2A). In P5, treatment with A0.1-A10 enhanced cell perimeter (Fig. 2B).

Aspect ratio (AR) describes the proportional relationship between the width of a cell and its length. P10 in A0-A0.5 groups had lower AR but P10 in A5-A10 groups showed higher AR compared with P5. P5 treated with A0.5-A10 significantly different from P5A0,

whereas P15 treated with A5 notably had higher AR than P15A0 (Fig. 2C). Roundness demonstrated the opposite trend from AR (Fig. 2D).

Circularity is a ratio of area and perimeter. The value of one means that the object has a circular shape. Solidity differentiates the convex and the concave cell area (*Hart et al., 2017*). Alteration in circularity and solidity implies changes in cell deformability and cell shape (*Pasqualato et al., 2013*; *Hart et al., 2017*). P5A0 had the highest values of circularity and solidity (Figs. 2E and 2F). Circularity of most P10 and P15 significantly dropped compared with P5 in their respective treatment groups. P5 cells incubated with ALN at every concentration showed lower circularity than P5A0, whereas P10 and P15 cells treated with ALN at moderate concentrations showed reduction in circularity than P10A0 and P15A0, respectively (Fig. 2E). Solidity showed a similar trend as circularity but in a lesser extent (Fig. 2F).

## Alteration of nuclear morphology in early, extended, and late passages under ALN-free and ALN conditions

Nuclei of the cells presented in bright blue color. The shape and size of P5A0 nuclei were homogenous and had oval shape (Fig. 3Ai). Some nuclei of P10A0 and P15A0 appeared larger and less consistent (Figs. 3Bi and 3Ci). Nuclear fragmentation was observed as shown in inset of Fig. 3Ci. ALN treatment caused uneven nuclear shape and size of some nuclei. Nuclear fragmentation was also monitored (arrows) (Fig. 3).

The numbers of nuclei represented the numbers of cells grown on the well plate. P5A0 displayed the highest value of nuclei. P10A0 and P15A0 nuclear numbers drastically declined compared with P5A0, implying slower growth rate in higher passages. The same pattern was seen in all ALN-treated groups except for A10 group. Every ALN concentration reduced the numbers of nuclei in P5 group, while moderate concentrations decreased nuclear numbers in P10 and P15 groups (Fig. 4A).

In every ALN group, nuclear area and perimeter of P5 were smaller than those of P10 and P15. Nuclear area and perimeter of P5 were smaller in response to ALN at lower concentrations, whereas those of P10 and P15 changed at higher concentrations (Figs. 4B and 4C).

AR values opposed to roundness values (Figs. 4D and 4E). In A0-A0.5 groups, P10 and P15 had lower AR but higher roundness, suggesting rounder shape nuclei. On the other hand, AR and roundness of P10 and P15 in A5-A10 groups illustrated less circular shape nuclei compared with P5. DPSCs at P5 and P15 subjected to ALN treatment showed a reduction in AR and an increase in roundness (Figs. 4D and 4E).

In general, circularity and solidity of P15 significantly reduced compared with those of P5 in every ALN group. Circularity and solidity of P15A10 were the lowest value in all condition (Figs. 4F and 4G). These values were related to the irregular shape of some nuclei and nuclear breakage (Fig. 3).

## Comparison of cell adhesion in early, extended, and late passages under ALN-free and ALN conditions

Crystal violet-positive cells after 5 h of cell seeding were shown in Fig. 5. The numbers of cell adhesion in P5 were significantly greater than P10 and P15 in untreated and treated

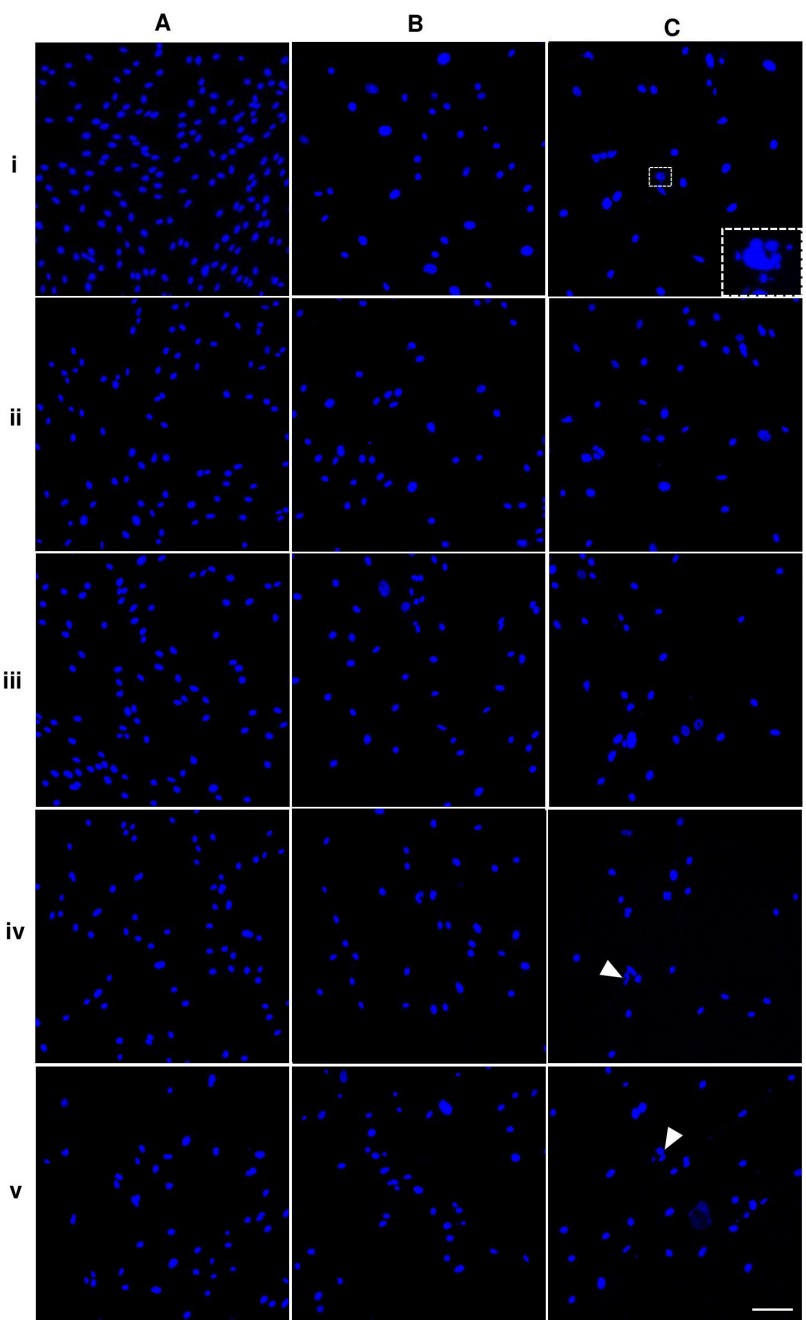

**Figure 3** **Nuclear morphology of dental pulp stem cells.**

groups (Fig. 6A). Cell area and perimeter of P15 were larger than those of P5 in ALN groups. DPSCs responded to ALN treatment by increased cell spreading (Figs. 6B and 6C).

In A0-A0.5 groups, higher passages had lower AR compared with P5. In P5 group, treatment with ALN decreased AR (Fig. 6D). Roundness had the opposite trend from AR (Fig. 6E).

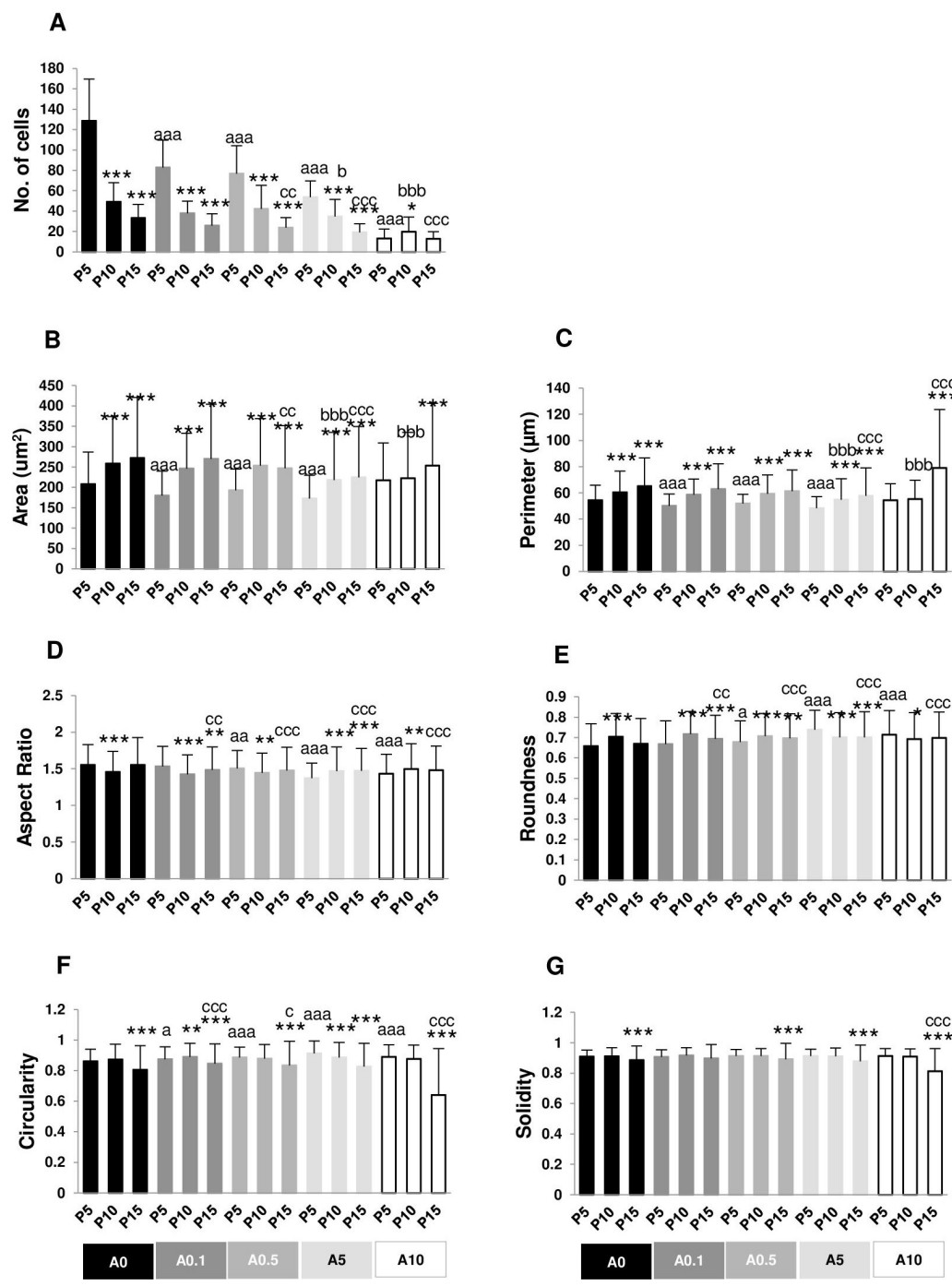

**Figure 4** (A–G) Measurement of nuclear morphology.

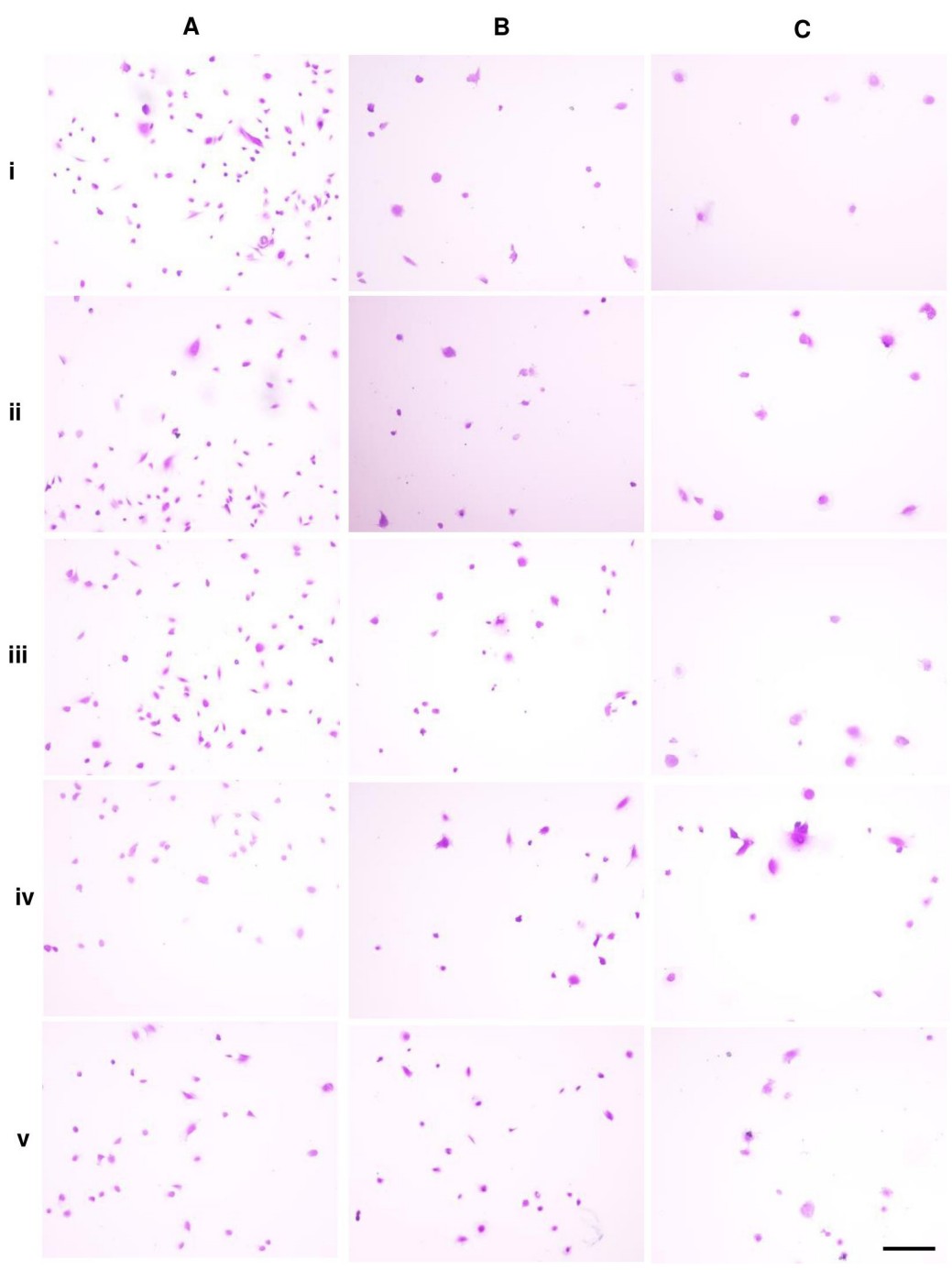

**Figure 5   Cell adhesion.**

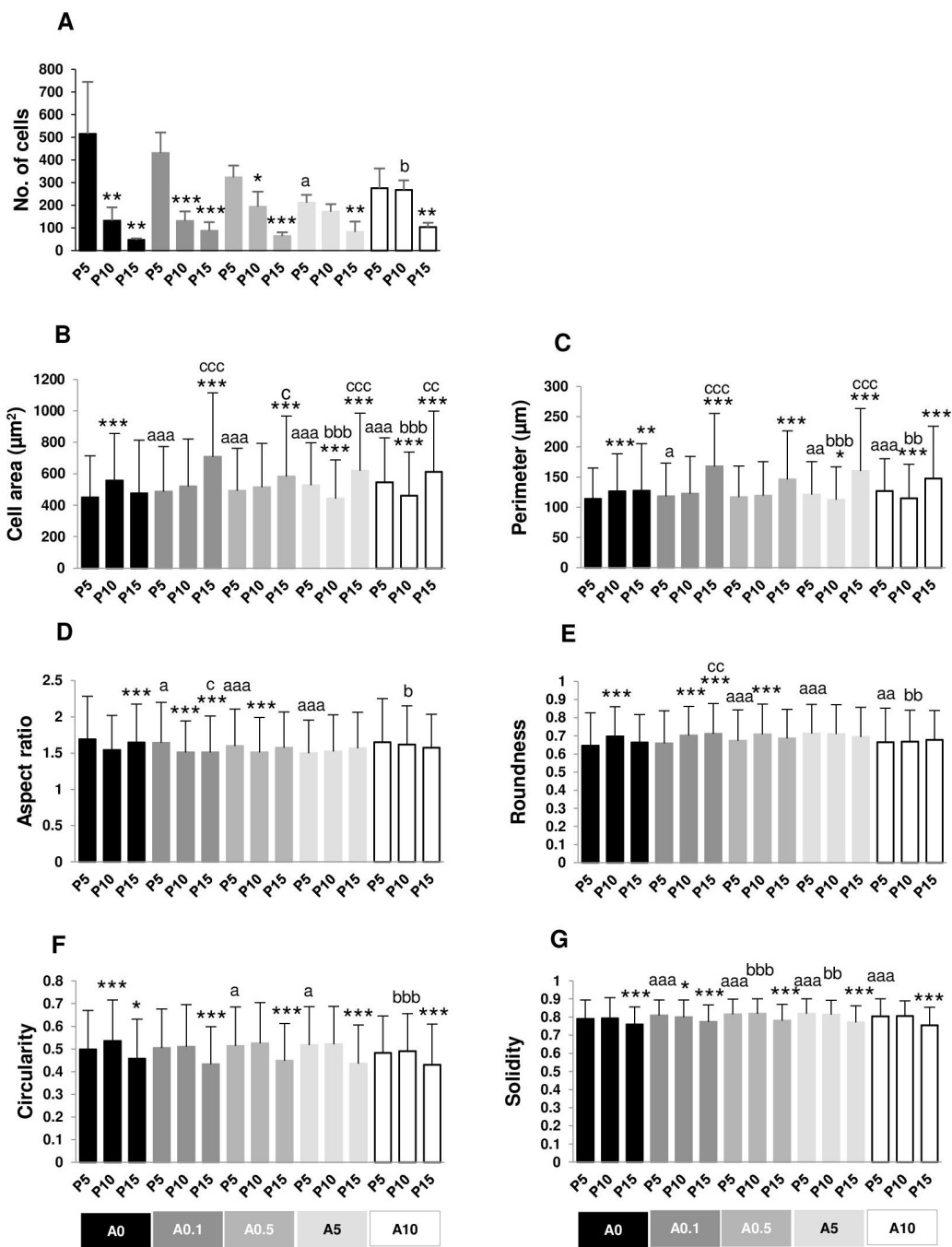

**Figure 6   (A–G) Measurement of cell adhesion.**

Cell circularity and solidity of P15 significantly decreased compared with those of P5 in every ALN condition. ALN treatment increased circularity and solidity of cells in P5 and P10 (Figs. 6F and 6G).

## Reduction of DPSC proliferative capability by replicative passaging and ALN addition

Figure 7 illustrates the proliferative capacity of DPSCs. Without ALN, cells in each passage had normal growth curve from day 1 to day 7. In A0 group, P5 cells maintained the optimal growth rate, whereas P15 had the slowest growth rate in all day tested. Compared with P5, P15 had the reduction rate approximately 50, 65, 70% for 1, 3, and 7 days, respectively (Fig. 7). In ALN treatment groups, the proliferative rate of higher passages gradually reduced from P5. Low concentration of ALN did not affect cell proliferation. A5 and A10 significantly caused a considerable reduction in cell proliferation in every passage compared with their respective passages. Long-term treatment and moderate dose of ALN almost abolished cell proliferation (Fig. 7).

## Effects of cell passages and ALN on total protein and ALP activity

The total protein of DPSCs cultured in OM for 7 days was extracted and measured. In general, P5 had higher total protein than P10 and P15 in each ALN group. P10 and P15 significantly had lower total protein compared with P5 in the presence of A5 and A10 (Fig. 8A).

ALP activity was obtained from cell lysate and the release into the media. ALP of P10 significantly dropped compared with that of P5 in A0, A0.1, and A0.5 groups, while ALP of P10 increased compared with that of P5 in A10 groups (Fig. 8B). ALN affected P5 cells by reducing ALP activity in a dose-dependent manner. ALP of P10A5 was enhanced compared with that of P10A0 (Fig. 8B). No significant change was observed in ALP release in the media in all conditions (Fig. 8C).

## Effects of cell passages and ALN on gene expressions

Since DPSCs can differentiate into odontoblastic or osteoblastic cells, genes such as Col I, OC, BSP, DSPP, and DMP1 were examined. The data set were presented in 2 aspects: within the same ALN treatment group and within the same passage group. P15 had higher Col I gene expressions than P5 in A0.5, A5, and A10 groups. On the contrary, P15 and P10 had lower OC gene expressions than P5 in A0.5 and A10 groups, respectively. There was no change in BSP, DSPP, and DMP1 genes (Fig. 9A). In P10 and P15, only A10 downregulated Col I gene expressions compared with A0. There was no change in other genes (Fig. 9B).

## DISCUSSION

DPSCs are MSCs that have displayed multi-differentiation potential toward odontoblastic and osteogenic cells (*Mangano et al., 2011*; *Tatullo et al., 2015*; *Rodas-Junco & Villicana, 2017*; *Sushmita et al., 2019*). These cells have become valuable alternative source of cells for the use in MSC-based therapies and studies varied from *in vitro* to *in vivo* (*Monti et al., 2017*; *Sushmita et al., 2019*). To obtain sufficient number of cells, continuously passaging primary cells can gradually lead to genetic and phenotypic changes, which could affect

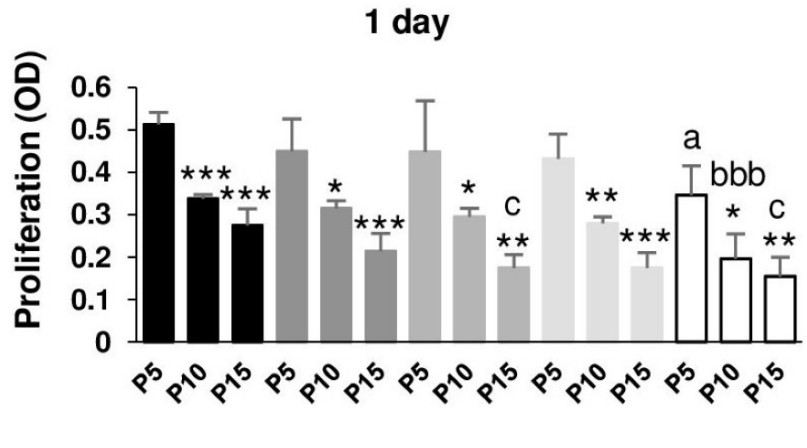

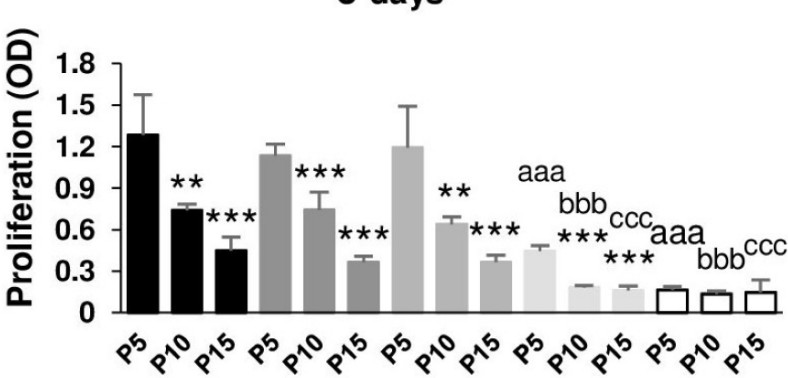

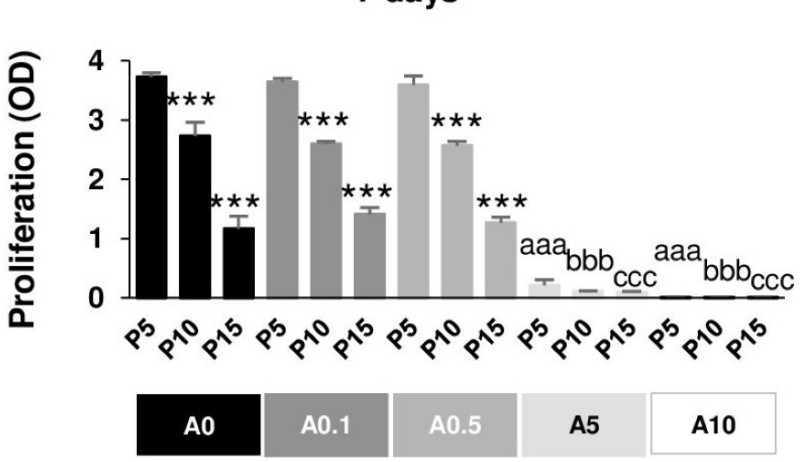

**Figure 7 Cell proliferation.**

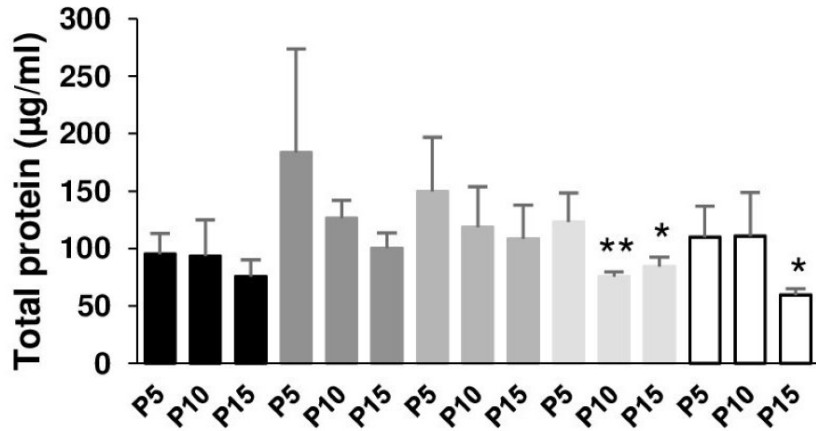

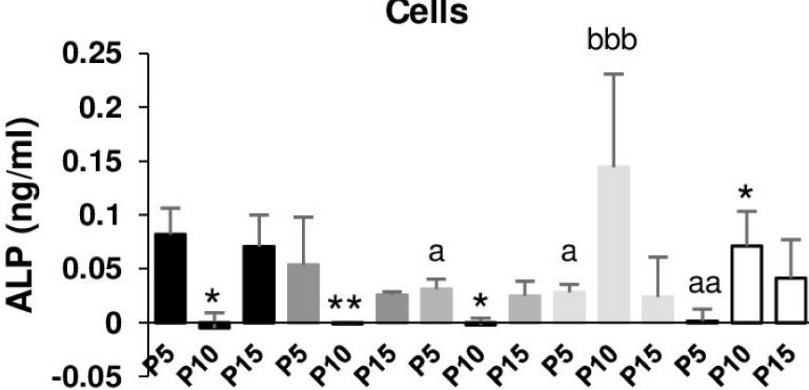

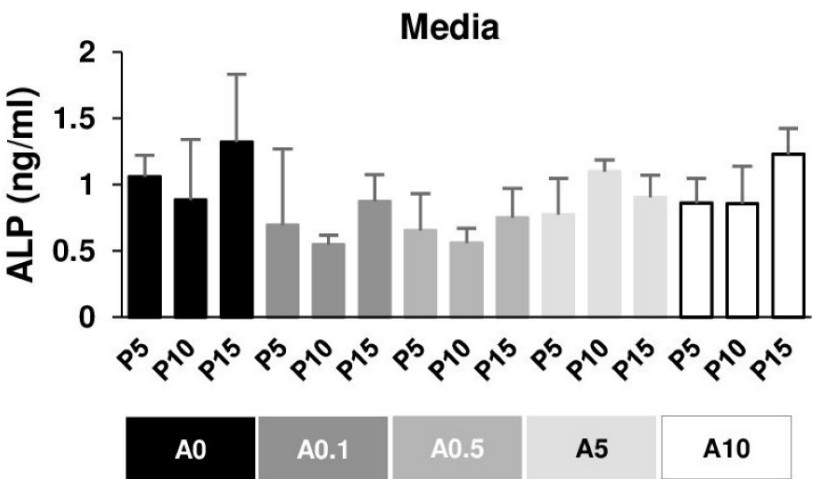

**Figure 8  Total protein and alkaline phosphatase activity.**

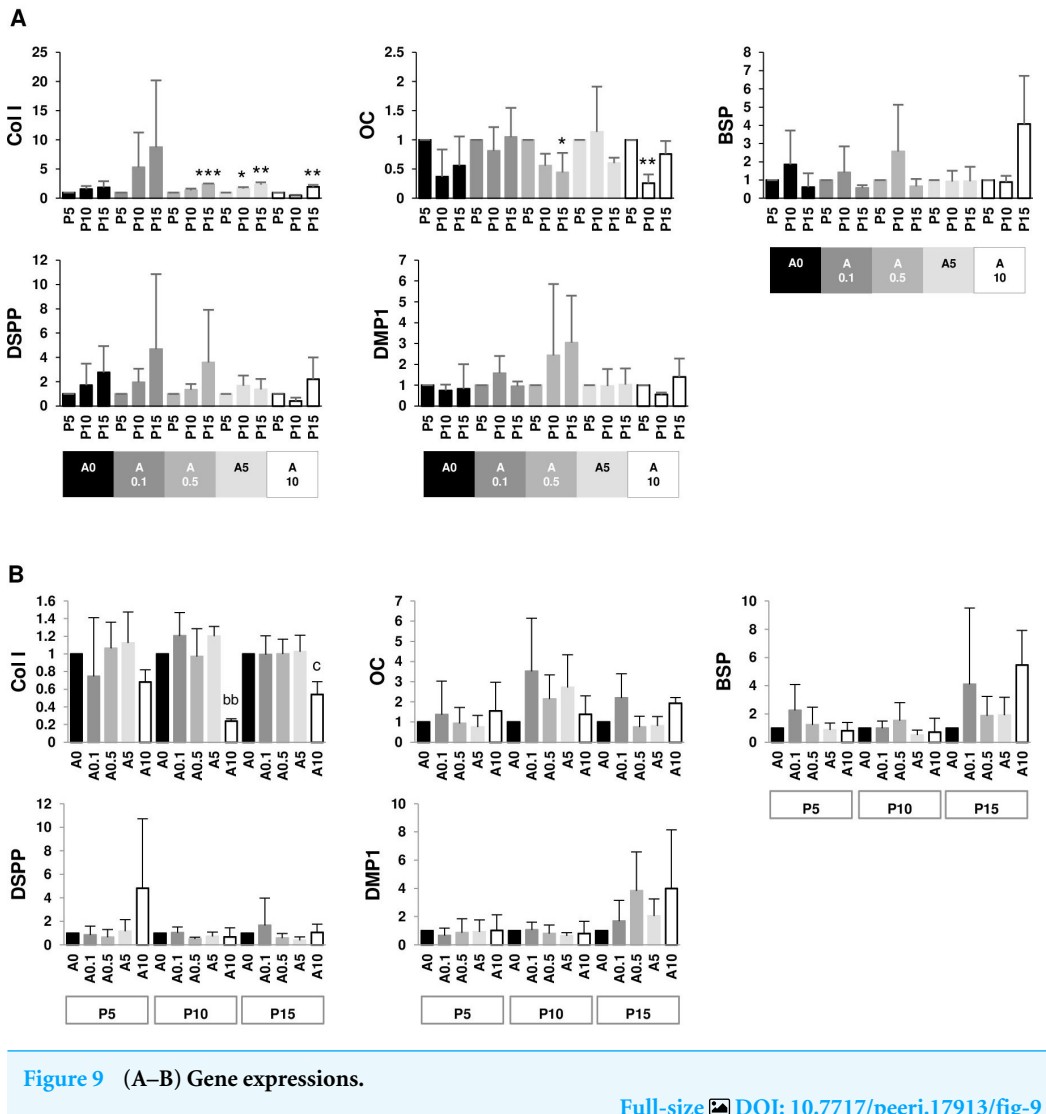

**Figure 9** (A–B) Gene expressions.

the use and the results in the experiments. The data from this study contributed that continuous cell expansion affected the experimental outcomes such as cell shape, activities as well as the response of cells to the ALN drug treatment (Summary shown in Fig. 10). After cell seeding, cells adhere to the substrate by making contact with the substrates, then spreading, and increasing their contact radius. The peak cell radius is observed at 18 h post-incubation (*Fritsche et al., 2013*). The parameters that comprehensively and accurately reflect the process of cell attachment and spreading include cell number, cell area and area fraction, relative and accumulative frequency of cell area, cell circularity, perimeter, and Feret's diameter (*Wang, Guo & Zhang, 2021*). We demonstrated for the first time that DPSC adhesion and its shape were influenced by cell passages and ALN addition. ALN has been shown to affect pre-osteoblast adhesion at high passages by decreasing cell adhering to the titanium surface (*Lilakhunakon, Suwanpateeb & Patntirapong, 2021*). The quantitative assessment of cell shape helps elucidate the mechanism of initial cell adhesion,

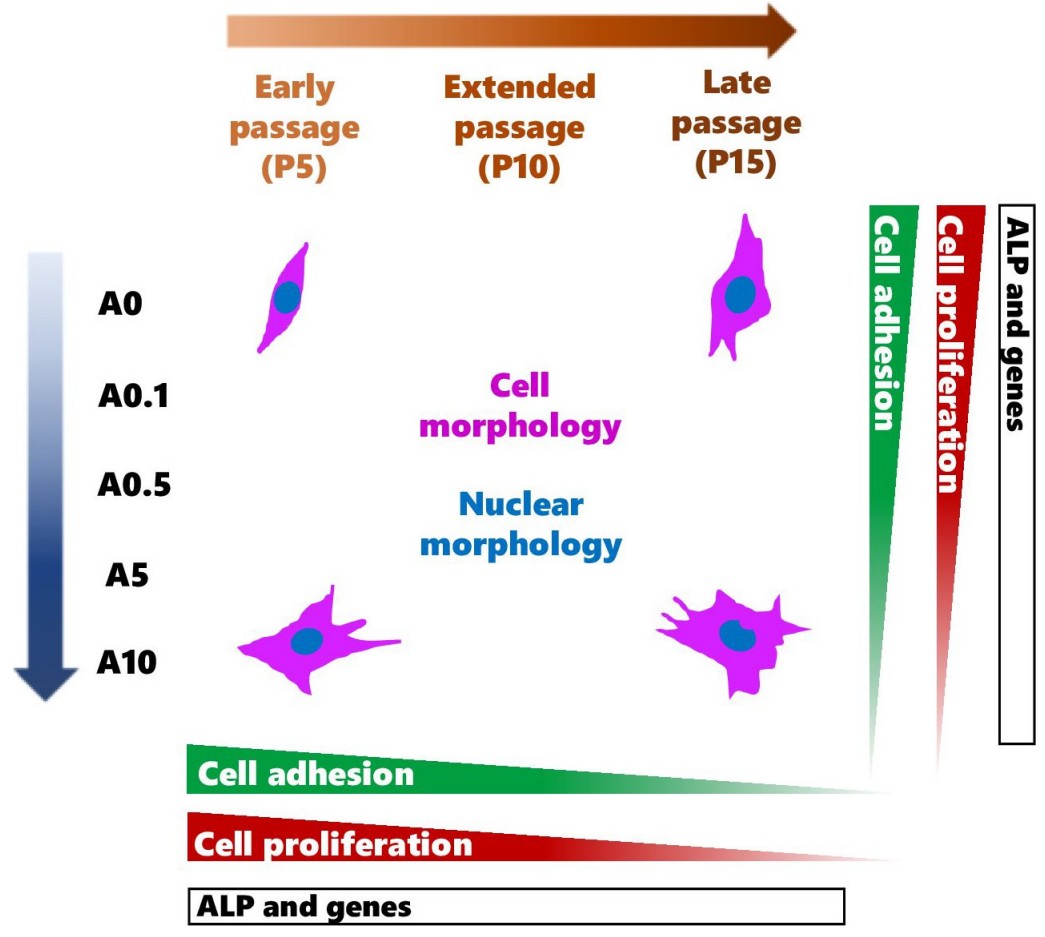

**Figure 10 A schematic diagram summarizes the effects of continuous cell passaging and ALN on cell morphology, nuclear morphology, cell adhesion, cell proliferation, and ALP activity.**

thus relatively estimating the direct interaction between cells and the substrate surface. Changes in cell shape by cell passages and ALN could exhibit the alteration of DPSCs and the substrate surface interaction. Hence, the use of cells for regenerative medicine might be limited to lower passage especially at the presence of ALN.

Late passages of DPSCs had larger cell size, heterogenous uniform, and increase in cytoplasmic granularity as previously observed in DPSCs and another MSCs *in vitro* and *ex vivo* (*Madeira et al., 2012*; *Oja et al., 2018*; *Wang et al., 2018*; *Mammoto et al., 2019*). Cell size of bone marrow MSCs is visibly enlarged resulting in a 4.8-fold increase at P6–9 as compared to P1 (*Oja et al., 2018*). *Ex vivo* endothelial cells isolated from small blood vessels in adipose tissues show age-dependent increases in cell size (*Mammoto et al., 2019*). Furthermore, change in cell area is correlated with biochemical senescence markers such as p16[INK4a] expression and senescence-associated $\beta$-galactosidase activity, suggesting a typical characteristic of aging cell (*Oja et al., 2018*). In this study, moderate dose of ALN also increased cell size of DPSCs. The replicative aging and ALN cause cell cycle arrest (*Patntirapong et al., 2021b*; *Sanagawa et al., 2022*). An accumulation of the cells in the

G2/M phase delay cells to enter into mitosis (*Patntirapong et al., 2021b*), thus increasing in the size of cells. Nuclear morphology, which is served as an indicator of cellular aging, shows a larger size in cultured aging cells and replicative senescent cells (*Heckenbach et al., 2022*; *Hartmann et al., 2023*). Nuclear area of DPSCs demonstrated a larger size in serial expansion but was smaller when receiving ALN. Cell and nuclear area monitoring can be applied to many types of cell culture systems (*Oja et al., 2018*) as well as could be used for routine detection and prediction of mesenchymal cell aging and abnormality under an inverted microscope.

Cell shape change can be distinguished by the alteration of cell shape descriptors such as circularity and solidity parameters (*Patntirapong, 2023*). Circularity and solidity values indicate cell deformability and the presence of membrane protrusions including lamellipodia, filopodia, and blebbing. High values suggest lower cell deformability and fewer protrusions (*Patntirapong, 2023*). The present data exhibited the reduction of cell circularity and solidity in continuous expansion and ALN treatment, implying higher cell deformability and more cell protrusions. Furthermore, nuclear shape of replicative senescent cells is irregular (*Heckenbach et al., 2022*). Circularity and solidity values of nuclei together with nuclear irregular shape and nuclear breakage indicated abnormal nuclei after long-term subculture and/or obtaining moderate dose of ALN. Late passage cells with or without ALN treatment could drive the cells into cell death.

One of the properties of stem cells is an ability to proliferate. DPSC proliferation was passage dependent, which gradually reduced in increasing passage numbers. Although the rate of cell growth slowed down significantly compared with their early passage counterpart, late passage of DPSCs still proliferated. A reduction in the proliferative capacity of P15 did not yet reach the Hayflick limit but showed sign of replicative aging according to Ogrodnik (*Ogrodnik, 2021*). The optimal proliferative capacity of DPSCs is reported at around P9 (*Martin-Piedra et al., 2013*) and still have high cell viability, functionality, and intact membrane integrity up to P14 (*Martin-Piedra et al., 2014*). The proliferation rate is reduced in late passage because the population doubling time in the late passage is longer than that in early passage (P9 at 3.42 days *vs.* P1 at 1.83 days) (*Yu et al., 2010*). Cells beyond P14 show a degree of cell membrane damage associated with metabolic impairment, suggesting a pre-apoptotic process (*Martin-Piedra et al., 2014*). The decrease in proliferative ability *in vitro* was consistent with decreased proliferative ability in *ex vivo* aged donors. The stem cells derived from young donors (up to 25 years) maintain proliferative ability in all cell passages tested. Cells from the aged group (up to 67 years) demonstrate a decline in proliferative ability (*Bressan et al., 2012*). The addition of ALN alone reduced cell proliferation in every cell passage tested. The presence of ALN stimuli might cause some cells to undergo premature programmed cell death earlier than others since ALN can trigger cell cycle arrest and cell damage (*Patntirapong et al., 2021b*). The combined effects of drug treatment and cell aging synergistically inhibited cell growth.

Differentiating DPSCs at different passages responded to stimuli differently. ALP is one of osteogenic/odontogenic markers. DPSCs at P5 and P10 responded in a different direction under ALN stimuli. P5 cells reacted to ALN treatment by reducing ALP levels in a dose-dependent manner, while P10 enhanced ALP level under ALN treatment. P10 had

lower ALP activity in untreated and low dose ALN conditions but produced more ALP activity after receiving moderate dose ALN. DPSCs at P9 under OM present a higher ALP level than DPSC at P1, suggesting that a more advanced passage DPSCs are more inclined toward osteogenic/odontogenic lineage (*Yu et al., 2010*). This study did not show the same trend as previous report (*Yu et al., 2010*). Different responses of cells to cell passages and ALN were also observed at the gene levels. In ALN-free condition, osteogenic/odontogenic genes did not change by cell passage. Under ALN, Col I gene expressions increased, whereas OC gene expressions decreased in P15. However, Col I gene expressions were reduced by ALN. It has been shown that late passage of DPSCs exhibits lower osteogenic genes (*Wang et al., 2018*). The passage used, genes tested, and the experimental settings might play a role in the different results. Since the results are inconsistent, more research such as the bone formation assay would be essential to further confirm the effects of cell passages and ALN on cell differentiation.

MSC populations including DPSCs are able to expand *ex vivo/in vitro* for several passages. Nevertheless, cells cultured over a long period will eventually lose their fitness to the point where cells are compromised and insufficient to support long-term use. Late passage underwent alteration from its original characteristics at earlier passages, as observed by changes in all parameter tested. These data were in accordance with previous reports (*Martin-Piedra et al., 2013*; *Martin-Piedra et al., 2014*; *Wang et al., 2018*; *Abdik et al., 2019*; *Mammoto et al., 2019*; *Heckenbach et al., 2022*). It has been suggested that primary cells at a passage of less than 10 might be optimal for studies and tissue engineering purpose because these cells still have adequate qualities (*Martin-Piedra et al., 2013*; *Liao et al., 2014*). Cells higher than P14 do not fulfill the quality control requirements and is recommended to be discarded (*Martin-Piedra et al., 2014*) to minimize the risk of losing their stemness capacity (*Lizier et al., 2012*) and avoid the changes in phenotypic and genetic properties (*Liao et al., 2014*; *Martin-Piedra et al., 2014*; *Wang et al., 2018*) as well as to avoid susceptibility to the microbial contamination.

Replicative passaging demonstrates changes in the function of transporters in cells, thus altering cellular uptake of the substrate (*Sanagawa et al., 2022*). Difference in cellular uptake could direct cell response to external factor differently and caused variable cell impairment in the presence of external factor. Based on the present results, cells lower than P10 is suggested for cell-based therapy or clinical studies under the presence of external factor. Cells at these passages would be able to sustain the use without undergoing significant alteration. Late passage would correspond to the studies of aged-related condition. The data in this study might provide guidance for the selection of appropriate and effective expanded DPSCs for distinctive study and therapeutic purposes. However, this study had a limitation. The biological responses of DPSCs were from one cell line. In the future, more lines of primary cells derived from various donors should be conducted to strengthen the data.

## CONCLUSION

Long-term subculture and ALN addition modulated DPSC behaviors at different extent *in vitro*. Without ALN condition, continuous cell expansion negatively affected number of cell

adhesion, proliferation, and differentiation markers. Late passage cells were heterogeneity and displayed one of antagonistic aging markers, which is morphological changes of cells. Early, extended, and late passages responded to ALN differently in most aspects of cell behaviors. It is necessary to understand several biological aspects of these dental stem cell populations. This is to ensure the potential and the extent of their efficacy to guarantee the success in each scientific purpose.

## ACKNOWLEDGEMENTS

This work was supported by Thammasat University Research Unit in Dental and Bone Substitute Biomaterials, Thammasat University. The authors also acknowledge Harikarn Mungpayabarn for technical assistance.

### Funding

This work was supported by Thammasat University Research Fund, Thammasat University, Contract No. TUFT 26/2566. The funders had no role in study design, data collection and analysis, decision to publish, or preparation of the manuscript.

### Grant Disclosures

The following grant information was disclosed by the authors:
Thammasat University Research Fund, Thammasat University: TUFT 26/2566.

### Competing Interests

The authors declare there are no competing interests.

### Author Contributions

- Somying Patntirapong conceived and designed the experiments, performed the experiments, analyzed the data, prepared figures and/or tables, authored or reviewed drafts of the article, and approved the final draft.
- Juthaluck Khankhow conceived and designed the experiments, performed the experiments, prepared figures and/or tables, and approved the final draft.
- Sikarin Julamorn conceived and designed the experiments, performed the experiments, prepared figures and/or tables, and approved the final draft.

### Data Availability

   Raw data are available as a Supplemental File.

### Supplemental Information

Supplemental information for this article can be found online at http://dx.doi.org/10.7717/peerj.17913#supplemental-information.

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
