# Peer review of "Long-term passage impacts human dental pulp stem cell activities and cell response to drug addition in vitro"

_PeerJ, doi:10.7717/peerj.17913_

## Round 0.1 · original submission · Minor Revisions

Dear authors,

Please refer to the reviewers' comments. Also know that it is a requirement of PeerJ to upload raw data, as relevant.

Reviewer 1 ·

Basic reporting

The manuscript clearly delivered the hypothesis, aim, and results. The discussion was also comprehensive to address a concern and opinion from the results. The overall writing is excellence.

Experimental design

Experimental design and methods are well organized and clearly stated in the manuscript.

Validity of the findings

No comment

Additional comments

The present study described the effect of long-term culture on dental pulp stem cell adhesion, proliferation, morphology and differentiation toward osteogenic lineage. The influence of ALN in these different DPSC passages was also examined.

The Rationale for study the effect of ALN should be clearly stated.

For the adhesion assay, the reason for evaluating at 5 hours must be stated. In other studies, cells can attach as early as 20 minutes after seeding.

One limitation of this study was the use of only one cell line purchased from Lonza. The observation observed may be only due to the biological responses of individuals. Using primary cell cultures derived from different donors could make the conclusion of these data more convincing. Please extensively discuss this point.

The mineralization assay is also suggested to further confirm the effect of passage and ALN on cell differentiation.

·

Basic reporting

This research is very interesting however it is suggested to add more explanation. The background is suggested to add to the reasons why researching ALN clearly and what influence ALN has on tooth development so that it is necessary to analyze influence of ALN on DSPC.

Some literature references are > 10 years old, it is suggested to replace with the latest references.

Professional article structure, images, tables but raw data isn't shared.

Experimental design

no comment

Validity of the findings

no comment

Additional comments

This study has described long-term subculture and addition of ALN modulates DPSC behavior at different extent in vitro.
In the discussion it is suggested add an explanation regarding the relationship between the results of this study and clinical implications, for example which passages are recommended for therapy or clinical research.

---

## Round 0.2 · accepted · Accept

Dear authors, I am happy to let you know that i am now accepting your manuscript for publication in PeerJ. Thank you.